# SARS-CoV-2 Symptoms during the Omicron Surge Differ between Boosted and Vaccinated Non-Boosted Persons

**DOI:** 10.3390/vaccines12030327

**Published:** 2024-03-19

**Authors:** Marisa A. Montecalvo, Paul Visintainer, Elizabeth Drugge, Katherine Kowalski, Rosemarie Raffa, Donna McKenna, Christine Moronta, Gary P. Wormser

**Affiliations:** 1Department of Medicine, Division of Infectious Diseases, New York Medical College, Valhalla, NY 10595, USA; gwormser@nymc.edu; 2Department of Health Services, New York Medical College, Valhalla, NY 10595, USA; katherine_kowalski@nymc.edu (K.K.); rraffa@nymc.edu (R.R.); donna_mckenna@nymc.edu (D.M.); cmoronta@nymc.edu (C.M.); 3Department of Epidemiology and Biostatistics, University of Massachusetts Chan Medical School–Baystate, Springfield, MA 01119, USA; paul.visintainer@baystatehealth.org; 4Department of Public Health, Division of Epidemiology, New York Medical College, Valhalla, NY 10595, USA; elizabeth_drugge@nymc.edu

**Keywords:** SARS-CoV-2, Omicron, symptoms, COVID-19 vaccination, boosters

## Abstract

**Purpose:** To determine the impact of booster COVID-19 vaccination on SARS-CoV-2 symptoms. **Background**: The Omicron surge of infections provided an opportunity to evaluate symptoms in relation to booster receipt. **Methods:** At a US medical college, the number, type, and duration of symptoms were evaluated for 476 students or employees, factoring in days between last vaccination and SARS-CoV-2 diagnosis. **Results:** Compared with vaccinated non-boosted individuals, boosted individuals reported a significantly higher frequency of nasal congestion (57.9% vs. 44.4%, *p* = 0.018) and nasal congestion and/or sore throat (77.2% vs. 62.0%, *p* = 0.003); in contrast, the frequency of body/muscle aches was significantly less among boosted individuals (22.1% vs. 32.4%, *p* = 0.038). With each one week increase in time since booster receipt, the probability of fever increased significantly by 4.4% (OR 1.044, 95% CI 1.01, 1.07, *p* = 0.001), and the probability of cough increased significantly by 4.8% (OR 1.048, 95% CI 1.01, 10.8, *p*= 0.010). **Conclusions:** Within a medical college population, during the first 7 months of the Omicron surge of infections, compared with vaccinated non-boosted individuals, boosted individuals significantly more often reported the following: nasal congestion as well as nasal congestion and/or sore throat. In contrast, body/muscle aches were reported significantly less often. The rates of fever and cough each significantly increased as time since booster dose receipt increased. These data suggest that having had a booster vaccination, as well the timing of receiving it, impacts the clinical manifestations of breakthrough SARS-CoV-2 infections. Additional studies are needed to precisely define SARS-CoV-2 symptoms in relation to booster vaccinations.

## 1. Introduction

After eleven months of morbidity, mortality, and social upheaval due to the SARS-CoV-2 pandemic, in December 2020, the emergency use authorization (EUA) of the mRNA COVID-19 vaccines brought great hope to the medical community. When the EUAs were issued, the vaccine efficacy for preventing symptomatic SARS-CoV-2 infection was 95% and 94.1% for the Pfizer-BioNTech and Moderna vaccines, respectively [1,2]. Authorization was based upon efficacy for the prevention of symptomatic infection, with a median follow-up of two months after vaccination. Breakthrough infections in vaccinated individuals, however, occurred and were seen with increasing frequency when the Delta variant emerged [3]. This was particularly well documented among healthcare workers who had the opportunity for regular testing [4,5]. By September 2021, based on evidence of waning immunity, together with the emergence of the more transmissible Delta variant, the Centers for Disease Control and Prevention (CDC) advised a booster vaccination for all individuals 65 years of age and older, long-term care facility residents, individuals aged 18–64 years old with risk factors for severe COVID-19, and individuals ages 18 years and older at higher risk of exposure due to their occupation [6].

When the Omicron variant first emerged in November 2021, it appeared that Omicron infections had a better outcome compared with earlier variants [7], and this proved to be a consistent finding [8,9]. However, the rapid rise of breakthrough infections in vaccinated individuals, and infections in individuals with a prior SARS-CoV-2 infection [10], demonstrated that both vaccine immunity and natural immunity offered less than optimal protection against Omicron infection. Data from the United Kingdom indicated that vaccine protection had decreased to less than 20% for individuals vaccinated approximately 5 months before exposure to the Omicron variant [11]. A booster dose of a mRNA vaccine, however, increased protection to 60% during the four-week period following booster vaccination [11,12,13]. This led public health officials to rely heavily upon booster vaccines to enhance immunity, and in December 2021, a booster dose was recommended for all individuals aged 18 years or older [14].

At New York Medical College (NYMC), all students and employees accessing the campus were required to have received a primary series of a COVID-19 vaccine and receipt of a booster when eligible. The purpose of this study was to determine if there were discernable differences in the clinical manifestations of breakthrough SARS-CoV-2 infections according to whether the person was vaccinated plus boosted versus vaccinated and not boosted.

## 2. Methods

This study evaluated SARS-CoV-2 infections occurring in NYMC students or employees from 1 December 2021 to 30 June 2022. NYMC is a health sciences college located in Westchester County, New York. Students were enrolled in the School of Medicine, the Touro College of Dental Medicine, the School of Health Sciences and Practices, physical therapy and speech and language pathology programs, and the Graduate School of Biomedical Sciences. The vaccination requirement was for all students and employees accessing the campus to have received a primary series of a COVID-19 vaccine (2 doses of mRNA vaccine or a single dose of the Janssen vector vaccine) by 30 September 2021, unless granted a medical or religious exemption to vaccination; on 15 February 2022, receipt of a booster COVID-19 vaccine when eligible was made a requirement.

All students and employees were required to report SARS-CoV-2 infections to the NYMC Health Services with a copy of the laboratory diagnostic test. Testing was required for all symptomatic individuals, asymptomatic individuals exposed to SARS-CoV-2, and individuals returning to NYMC from travel anywhere other than the states contiguous to New York State. Beginning in late December 2021, due to the lack of availability of laboratory diagnostic tests, a photograph of a positive home antigen test was accepted as confirmation of infection, and NYMC distributed home tests to students and staff, as needed, to ensure testing was readily available.

All SARS-CoV-2-confirmed cases were interviewed by Health Services staff, placed off-duty, and sent a letter delineating the New York State Department of Health (NYSDOH) requirements for isolation and provided a symptom log to complete and return to Health Services to be released from isolation and return to work or school. The symptom log began with the date of symptom onset, or the date of the positive test for asymptomatic individuals, and was completed for at least the number of days that isolation was required. On 5 January 2022, the NYSDOH shortened the minimum isolation requirement from ten days to five days if the person indicated that they were clearly improving by day five and were without fever for at least 24 hours [15]. Thus, all cases occurring on or after 1 January 2022 were subject to the five-day isolation requirement.

In accordance with the procedures of NYSDOH contact tracers, the symptom log required the recording of body temperature daily and to indicate daily whether fever, cough, or shortness of breath were present. Other symptoms were written on the log by the individual or were charted in the medical record through telephone and email correspondence. Symptoms were collated into ten primary categories, as defined in Table 1.

The number of primary symptoms was defined as the summation of each of the 10 primary symptom categories; the total number of symptoms was the summation of primary symptoms plus “other” if a symptom apart from the ten categories was recorded. The duration of symptoms was the number of days any symptom was present. The duration of isolation for symptomatic cases was the number of days from symptom onset until release from isolation. The date of infection was defined as the date of the positive SARS-CoV-2 diagnostic test. Cases were defined as boosted at the time of infection if a booster vaccine was received before the date of the positive test because CDC indicated that a two-week interval was needed to be considered “fully vaccinated” only after primary vaccination, but no timeframe was defined after booster receipt [16]. When NYMC required the receipt of a booster COVID-19 vaccine, this was in accordance with the CDC guidance at the time, which was the receipt of a booster dose five months after a mRNA vaccine series, or two months after a vector vaccine. One could defer a booster for three months after natural SARS-CoV-2 infection [17].

All symptom logs and medical records were reviewed twice by NYMC Health Services nurse practitioners. If there was a discrepancy between the two reviews, the case was reviewed a third time by the medical director in collaboration with a nurse practitioner. All data were entered into a standardized database that included demographics, the COVID-19 vaccinations received, SARS-CoV-2 infection within 6 months before the date of the SARS-CoV-2 infection during this study, and the following until release from isolation: specific symptoms, the number of days any symptoms were present, the number of symptoms reported, and the number of days in isolation.

### Statistical Methods

Data were collected in Excel and analyzed using Stata, v. 18 (Stata Corp, College Station, TX, USA). Descriptive statistics, including means and standard deviations (SDs) for continuous variables, and frequencies and proportions for categorical variables, were assessed. Differences between groups according to booster status were assessed using Student’s two-sample *t*-test for continuous variables and Fisher’s exact test for categorical variables. Logistic regression analysis was used to assess the change in the proportions of study subjects reporting particular symptoms over time since booster receipt (expressed as the change in the odds ratio for each increase of one week since booster receipt). Pearson’s correlation was used to assess the linear relationship between chronologic date of infection and time since booster receipt. A two-tailed significance level of 0.05 was used.

This study was approved by the Institutional Review Board of New York Medical College with a waiver for written informed consent.

## 3. Results

By 30 September 2021, 2443 of the 2456 (99.4%) NYMC students and employees had received a primary series of a COVID-19 vaccine; 13 individuals were granted a medical or religious exemption to vaccination. In December 2021, the incidence of SARS-CoV-2 infection at NYMC was over ten-fold greater than the incidence in each of the prior three months, co-incident with the establishment of Omicron (B.1.1.529) as the predominant variant causing infection in New York State (Figure 1). Between 1 December 2021 and 30 June 2022, there were 482 confirmed SARS-CoV-2 cases at NYMC, 6 of which occurred in unvaccinated individuals. The proportion of infections among unvaccinated individuals was significantly greater than the proportion among vaccinated individuals (46.2% vs. 19.5%, *p* = 0.016).

Unvaccinated individuals were not included in further analyses in this study. Of the 476 vaccinated cases, 347 (72.9%) were students and 129 (27.1%) were employees. Primary vaccination was by use of the Moderna mRNA vaccine for 246 (51.7%) cases, the Pfizer-N-Biotech mRNA vaccine for 211 (44.3%) cases, and the Janssen vector vaccine for 19 (4.0%) cases.

In total, 433 (90.9%) of the 476 cases were symptomatic, 34 (7.1%) cases were asymptomatic, and 9 (1.9%) cases reported an infection but without sufficient information to determine if the individual was symptomatic (Figure 2). A total of 393 (90.7%) of the 433 symptomatic cases had sufficient data to assess symptoms, symptom duration, and the duration of isolation; 40 cases were excluded due to insufficient data. Of the 393 SARS-CoV-2 cases assessable for symptoms, 285 (72.5%) were in boosted individuals and 108 (27.5%) in vaccinated, non-boosted individuals (Figure 2). A total of 6 (1.5%) of these 393 individuals also had a prior SARS-CoV-2 infection within six months of the infection date in this study. The vaccinated non-boosted cases occurred almost exclusively during December 2021 and January 2022 (Figure 1), in part due to the school requirement to receive a booster vaccination by 15 February 2022.

The 393 cases were in 389 individuals; 4 individuals suffered two infections. During the study period, cough, nasal congestion, and fever were the most common SARS-CoV-2 symptoms reported, followed by sore throat, headache, body/muscle aches, and fatigue. A change in, or loss of, taste or smell was only reported in thirteen cases (Table 1). Compared with individuals who were vaccinated and not boosted, boosted individuals reported a significantly greater frequency of nasal congestion (57.9% vs. 44.4%, *p* = 0.018), a significantly greater frequency of nasal congestion and/or sore throat, defined as having nasal congestion or sore throat or both (77.2% vs. 62.0%, *p* = 0.002), and a significantly lower frequency of body/muscle aches (22.1% vs. 32.4%, *p* = 0.038) (Table 2). Boosted individuals also isolated for significantly fewer days (mean (SD) 8.4 (2.9) days vs. 10.4 (2.9) days, *p* < 0.001) when the minimum isolation requirement was five days; 80.7% of cases in this study occurred during the five-day requirement period. However, although boosted individuals isolated for fewer days during the five-day isolation requirement period, no significant difference was found in the number of days symptoms were present or in the total number of symptoms reported.

Twelve boosted individuals had received the booster vaccine dose within 14 days of developing the SARS-CoV-2 infection. As these individuals may not have had a protective immune response by this time point, a supplemental analysis was performed classifying these cases as vaccinated non-boosted individuals. This change, however, did not impact the above discussed significant clinical differences that were observed between the boosted versus the vaccinated non-boosted groups (see Appendix A).

After 31 January 2022, during the remaining five months of the study, only five SARS-CoV-2 cases occurred in vaccinated non-boosted individuals. To better understand the differences in symptoms observed, a restricted analysis of cases occurring only from 1 December 2021 to 31 January 2022 was performed (Table 3). In this restricted timeframe, the boosted individuals had more recently received the booster vaccination, with a mean (SD) of 47.9 (30.6) days between booster receipt and date of infection. In this restricted group, boosted individuals reported a lower frequency of fever, but this difference was not statistically significant (38.5% vs. 51.5%, *p* = 0.098). Similarly, boosted individuals reported a higher frequency of nasal congestion and/or sore throat (75.6% vs. 63.1%, *p* = 0.078), but this difference was also not statistically significant.

Since the chronologic time frame of when infection occurred and the time since booster receipt could each potentially affect symptoms, the two variables were evaluated for correlation. The time interval from booster receipt to infection directly correlated with the chronologic time frame of the study, beginning in the first week of the study on 1 December 2021 and progressing through seven months until the study’s conclusion on 30 June 2022 (correlation co-efficient = 0.84) (Appendix A), which was expected since boosters were recommended at the beginning of this study (December 2021). When infected boosted cases were evaluated weekly, for each increase of one week since booster receipt, the probability of fever increased significantly by 4.4% (OR 1.044, 95% CI 1.01, 1.07, *p* = 0.001) and the probability of cough increased significantly by 4.8% (OR 1.048, 95% CI 1.01, 10.8, *p* = 0.010), whereas no significant correlations were found in the probability of nasal congestion and/or sore throat (*p* = 0.88), or in the probability of body/muscle aches (*p* = 0.10), (Figure 3), or for any of the other symptoms studied. 

## 4. Discussion

The school requirement to report all confirmed SARS-CoV-2 infections to Health Services and to report on symptoms to be released from isolation allowed us to study the clinical manifestations of SARS-CoV-2 infections early during the Omicron surge from the date of symptom onset to release from isolation and to compare vaccine-boosted versus vaccinated non-boosted individuals. Boosted individuals reported nasal congestion and nasal congestion and/or sore throat (nasal congestion or sore throat or both) significantly more often and reported body/muscle aches significantly less often than vaccinated non-boosted individuals. Although nasal congestion and sore throat [18,19,20,21,22], but without loss of taste or smell [22,23], have been associated with Omicron infection, to our knowledge, our findings have not been previously reported as distinguishing symptoms of boosted individuals compared with vaccinated non-boosted individuals. Only 1.5% of our cases had a natural SARS-CoV-2 infection within six months of the infection evaluated in this study, suggesting that the findings are truly associated with vaccine immunity. Our data suggest that some aspects of the booster-induced immune response may have promoted the development of nasal and pharyngeal symptoms while lessening certain systemic symptoms. When cases were restricted to the first two months of the Omicron surge of infections, boosted individuals reported a lower but not statistically significant frequency of fever, and importantly, we found that for each increase of one week since booster receipt, the probability of fever increased significantly, and the probability of cough also increased significantly. These changes (increased fever and increased cough) as booster immunity may be waning are consistent with a more severe infection. It is unclear why the symptoms associated with having a booster (more nasal congestion, more nasal congestion and/or sore throat, fewer body/muscle aches) did not change as time since booster receipt increased. This suggests that how waning immunity may affect symptom manifestations is complex, and other factors may also contribute to symptom manifestations, such as inoculum at site of infection or variant strain.

This study differs from others reporting symptoms of Omicron infection because we evaluated symptoms throughout the period of isolation, which was an average of 8.4–10.9 days depending upon the NYSDOH isolation requirement, whereas most prior studies collected symptom data often at a single point in time, such as the date of test positivity. Prior studies of Omicron symptoms have also had more diverse populations, such as any healthcare professional [18,19], individuals tested at community testing sites [20,21], and cases identified post-exposure [19,24] or during an outbreak [22], whereas our population included only students and employees at a single medical college. Prior studies comparing boosted with vaccinated non-boosted individuals have found that boosted individuals reported fewer systemic symptoms, such as fever, chills or myalgias [18,20,24], or fewer symptoms overall [25,26].

The HEROES-RECOVER Network [26], a large US database of healthcare personnel, first responders, and frontline workers who participate in self-testing weekly and report SARS-CoV-2 symptoms found that the number of symptoms reported for symptomatic cases that were vaccinated and boosted 7–149 days before infection was less than the number of symptoms reported by those who had received two vaccine doses at least 150 days prior to infection (but were not boosted): mean (SD) 5.9 (3.2) symptoms for boosted cases vs. 7.3 (3.5) symptoms for vaccinated non-boosted cases. Of interest, in the HEROES-RECOVER Network study the vaccinated and boosted cases were more likely to develop a symptomatic SARS-CoV-2 infection than those who were entirely unvaccinated (OR, 2.0 [95% CI: 1.1–3.5]). Although we did not find a significant difference in the number of symptoms or in the duration of symptoms for boosted individuals compared with vaccinated non-boosted individuals, in our study, boosted individuals isolated for significantly fewer days when the isolation requirement was five days. However, we suspect that there were un-measurable factors that influenced the duration of symptom monitoring and isolation. Students were significantly more likely to have received a booster vaccine compared with employees, and perhaps students felt an urgency to return to school to not miss clinical activities, as had occurred earlier during the pandemic [27], and this may have influenced symptom reporting.

There are several limitations to our study. The symptom log used only asked specifically about the daily presence of fever, cough, and shortness of breath; all other symptoms were compiled based on what was written on the log, or by telephone or email correspondence. Second, vaccinated non-boosted individuals were infected almost exclusively during the first two months of the study. Third, although the timeframe of when infection occurred during the study could have influenced symptoms, as Omicron subvariants came into circulation [28], the timeframe of infection date was highly directly correlated with time since booster receipt, and a linear regression analysis was able to demonstrate how time since booster receipt significantly impacted certain symptoms. Fourth, no specific data were collected on the use of antiviral drugs or on the presence of co-morbidities. However, we know from general health services data that co-morbidities are limited in the student population. We also think that the use of antiviral drugs was limited because oral antivirals were indicated for emergency use only for individuals at high risk of severe COVID-19. Our population is not at risk for severe disease based upon age, and we know from NYMC Health Services data that fewer than 0.4% of infections have resulted in hospitalization (and these hospitalizations occurred before the Omicron variant emerged). Fifth, we assume that our findings are related to infection caused by the Omicron variant based upon the timeframe, which was coincident with the extremely rapid increase in SARS-CoV-2 infections during the Omicron surge, but no testing was performed to verify this. Lastly, findings from this well-defined medical college population may not be generalizable to larger more diverse populations.

## 5. Conclusions

Our findings that boosted individuals had a significantly higher frequency of nasal congestion and of nasal congestion and/or sore throat but significantly lower frequency of body/muscle aches, and that both the frequency of fever and the frequency of cough were directly associated with increased time since booster receipt, have important implications for booster recipients. Although COVID-19 vaccination clearly provides protection against hospitalization and death, and booster vaccination increases that protection [29,30], based on our study results, future efficacy studies should also address the frequency of particular symptoms in breakthrough infections, as well as the number of symptoms, symptom severity, and symptom duration.

## Figures and Tables

**Figure 1 vaccines-12-00327-f001:**
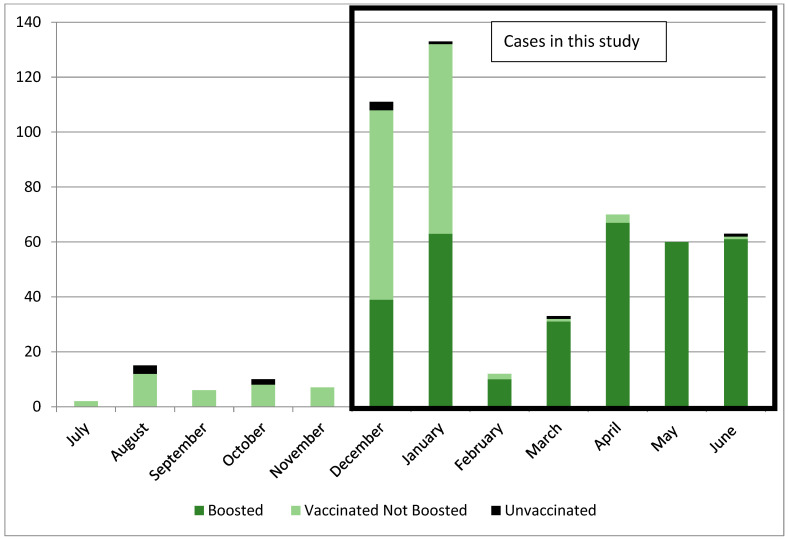
SARS-CoV-2 infections according to vaccination status from July 2021 to June 2022. The cases included in this study are those cases occurring from December 2021 to June 2022.

**Figure 2 vaccines-12-00327-f002:**
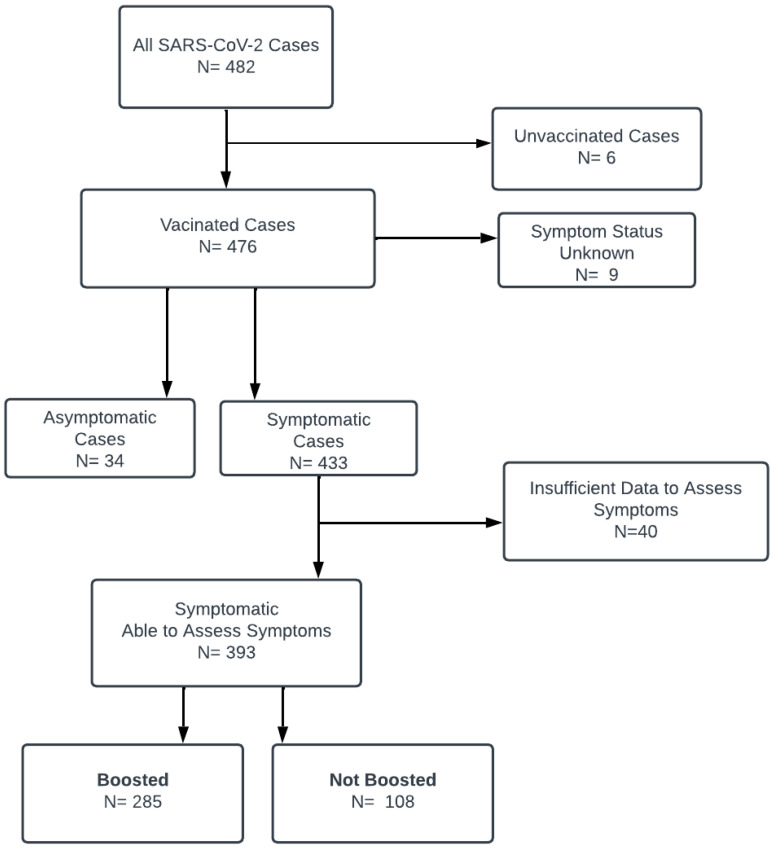
Cases of SARS-CoV-2 infection according to vaccination status, symptoms, and whether symptoms could be assessed.

**Figure 3 vaccines-12-00327-f003:**
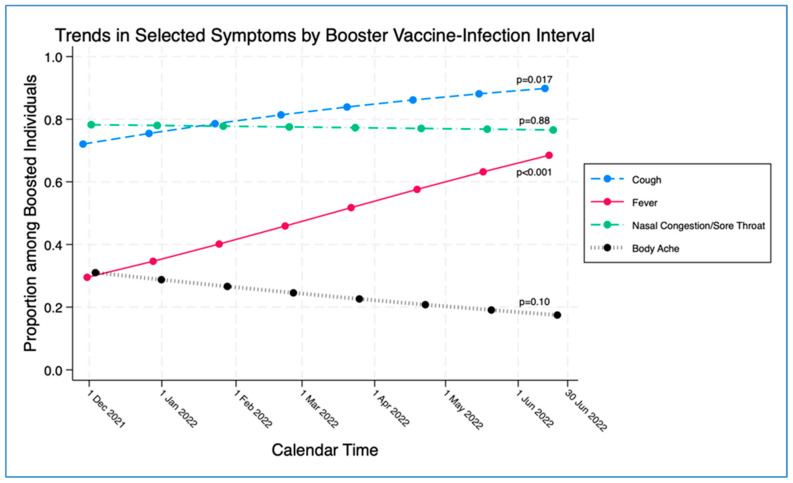
The trend in proportions estimated by the logistic regression model of vaccinated and boosted individuals with symptoms of cough, fever, nasal congestion and/or sore throat, and body/muscle aches according to the number of weeks from receipt of booster vaccination until date of SARS-CoV-2 infection.

**Table 1 vaccines-12-00327-t001:** Symptoms of SARS-CoV-2 Infections Reported.

Symptom	Definition/Terms Used to Capture Symptom	Number (%) Cases N = 393
**Symptoms Specifically Requested on Symptom Log**		
*Fever*	*Temp ≥ 100° Fahrenheit or “fever” recorded on log*	211 (53.6)
*Cough*	*Cough, chest congestion*	327 (83.2)
*Shortness of breath*	*Shortness of breath*	55 (13.9)
**Symptoms Not Specifically Requested on Symptom Log**		
Nasal congestion	Nasal congestion, runny nose, stuffy nose, sneezing, post nasal drip, head congestion, head cold	213 (54.1)
Sore throat/pharyngeal	Sore throat, scratchy throat, itchy throat, tingling in throat, dry throat, pharyngitis, tonsillitis, swollen tonsils	183 (46.5)
Headache	Headache, migraine	110 (27.9)
Fatigue	Fatigue, lethargy, tired, exhausted	100 (25.4)
Body/muscle aches	Body aches, muscle aches, myalgias, achy, back pain, leg pain, joint pain	98 (24.9)
Gastrointestinal	Nausea, vomiting, diarrhea, stomach upset, stomach cramping, abdominal pain	28 (7.1)
Change in taste or smell	Loss of taste, loss of smell, change in taste or smell	13 (3.3)
Other *		73 (18.5)

* Seventy three cases reported 83 symptoms as follows: sinusitis or sinus pressure/pain n = 11; chills or sweats without fever n = 10; neurologic symptoms (head fog, dizziness, light-headedness, vertigo) n = 10; malaise n = 9; weakness or muscle weakness n = 9; laryngitis/hoarseness n = 6; chest symptoms (chest pain, pleural pain, chest tightness) n = 5; non-specific viral symptoms n = 5; and 18 cases with a complaint that was reported in four or fewer cases.

**Table 2 vaccines-12-00327-t002:** Characteristics of individuals and symptoms reported according to vaccination status.

	Not Boosted	Boosted	*p* Value
	N = 108	N = 285	
**Gender = Female, No. (%)**	64 (59.3)	163 (57.2)	0.733
**Age in years, Mean (SD) ***	31.6 (10.7)	31.3 (12.1)	0.813
**Student versus Employee Status, No. (%)**			0.032
Student	69 (63.9)	214 (75.1)	
Employee	39 (36.1)	71 (24.9)	
**Days Since Last Vaccine Until Infection, Mean (SD)**	244.9 (69.2)	123.7 (64.4)	<0.001
**Primary Symptoms Present, No. (%)**			
Cough	87 (80.6)	240 (84.2)	0.450
Fever—present or temperature ≥ 100 °F	57 (52.8)	154 (54.0)	0.822
Nasal congestion	48 (44.4)	165 (57.9)	0.018
Sore throat	44 (40.7)	138 (48.4)	0.176
Headache	37 (34.3)	73 (25.6)	0.102
Body/muscle aches	35 (32.4)	63 (22.1)	0.038
Fatigue	23 (21.3)	77 (27.0)	0.299
Shortness of breath	21 (19.4)	34 (11.9)	0.072
Gastrointestinal symptoms	10 (9.3)	18 (6.3)	0.379
Change in taste or smell	7 (6.5)	6 (2.1)	0.052
Nasal congestion and/or sore throat	67 (62.0)	220 (77.2)	0.003
**Number of Primary Symptoms Reported, Mean (SD)**	3.4 (1.8)	3.4 (1.5)	0.911
Other symptoms, No. (%)			0.427
0	95 (88.0)	245 (86.0)	
1	10 (9.3)	36 (12.6)	
2	3 (2.8)	4 (1.4)	
**Number of Primary Plus Other Symptoms, Mean (SD)**	3.6 (1.9)	3.6 (1.6)	0.943
**Days Symptoms Present, Mean (SD)**	7.1 (3.2)	6.3 (2.9)	0.014
Cases before 1 January, required isolation = 10 days n = 76	7.5 (3.1)	6.3 (2.5)	0.112
Cases on/after 1 January, required isolation = 5 days n = 317	6.8 (3.1)	6.3 (2.8)	0.248

**Days Isolated, Mean (SD)**			
Cases before 1 January, required isolation = 10 days n = 76	10.9 (1.6)	11.1 (1.7)	0.661
Cases on/after 1 January, required isolation = 5 days n = 317	10.4 (2.9)	8.4 (2.9)	< 0.001

* SD = standard deviation.

**Table 3 vaccines-12-00327-t003:** Characteristics of individuals and symptoms reported according to vaccination status for cases occurring early during the Omicron surge: December 2021–January 2022.

	Not Boosted	Boosted	*p* Value
	N = 103	N = 78	
**Gender = Female, No. (%)**	60 (58.3)	47 (60.3)	0.879
**Age in years, Mean (SD) ***	31.6 (10.8)	32.1 (12.1)	0.778
**Student versus Employee Status, No. (%)**			0.421
Student	67 (65.0)	56 (71.8)	
Employee	36 (35.0)	22 (28.2)	
**Days Since Last Vaccine Until Infection, Mean (SD)**	244.9 (68.6)	47.9 (30.6)	<0.001
**Primary Symptoms Present, No. (%)**			
Cough	84 (81.6)	60 (76.9)	0.462
Fever—present or temperature ≥ 100 °F	53 (51.5)	30 (38.5)	0.098
Nasal congestion	46 (44.7)	40 (51.3)	0.453
Sore throat	43 (41.7)	37 (47.4)	0.455
Headache	36 (35.0)	23 (29.5)	0.522
Body/muscle aches	33 (32.0)	22 (28.2)	0.627
Fatigue	23 (22.3)	19 (24.4)	0.859
Shortness of breath	20 (19.4)	10 (12.8)	0.313
Gastrointestinal symptoms	9 (8.7)	6 (7.7)	1.000
Change in taste or smell	7 (6.8)	1 (1.3)	0.140
Nasal congestion and/or sore throat	65 (63.1)	59 (75.6)	0.078
**Number of Primary Symptoms Reported, Mean (SD)**	3.4 (1.8)	3.2 (1.6)	0.314
Other symptoms, No. (%)			0.861
0	90 (87.4)	69 (88.5)	
1	10 (9.7)	8 (10.3)	
2	3 (2.9)	1 (1.3)	
**Number of Primary Plus Other Symptoms, Mean (SD)**	3.6 (2.0)	3.3 (1.7)	0.308
**Days Symptoms Present, Mean (SD)**	7.3 (3.1)	6.7 (3.2)	0.261
Cases before 1 January, required isolation = 10 days n = 76	7.5 (3.1)	6.3 (2.5)	0.112
Cases on/after 1 January, required isolation = 5 days n = 105	7 (3.1)	6.8 (3.4)	0.861

**Days Isolated, Mean (SD)**			
Cases before 1 January, required isolation = 10 days n = 76	10.9 (1.6)	11.1 (1.7)	0.661
Cases on/after 1 January, required isolation = 5 days n = 105	10.7 (2.8)	9.8 (3.0)	0.102

* SD = standard deviation.

## Data Availability

Data are contained within the article.

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
