# Peer review of "SARS-CoV-2 Symptoms during the Omicron Surge Differ between Boosted and Vaccinated Non-Boosted Persons"

_vaccines, 2024, doi:10.3390/vaccines12030327_

Round 1

Reviewer 1 Report

Comments and Suggestions for Authors

This study examines disease outcomes in persons who were vaccinated against SARS-CoV-2 and later became infected with the Omicron variant. The study specifically aims to evaluate if those who received booster injections developed symptoms differently from those who only received the initial vaccine schedule.

Strengths of the study are the well-defined population at the New York Medical College and the in general well organized manuscript with clear tables

Weaknesses to be addressed:

  1. Although the population is well-define it is a very small population. This is a significant limitation and should be mentioned in the abstract.
  2. Given the well-defined population, information on how many got infected in the three groups (unvaccinated, vaccinated and not boosted, and vaccinated and boosted) is critical. Without this information, the manuscript may give the wrong impression of vaccine effectiveness. Specifically, of the entire NYMC population, how many were vaccinated and boosted in total, and what percentage of those got infected? Of the entire NYMC population, how many were vaccinated and not boosted in total, and what percentage of those got infected?
  3. As written the abstract kind of implies that being boosted is bad. Is that actually what you mean to say? Again, here the numbers for comment 2 above are critical.
  4. The conclusion section of the abstract does not mention that those not boosted had more body aches. This should be written more clearly.
  5. The abstract text containing the data ‘(77.2% vs. 62.0%, p= 0.002)’ doesn’t match Table 2. Furthermore, table 2 has the p value at 0.003. The text needs to be rewritten to better reflect what is compared and the p value checked and corrected in one place.
  6. Table 2 shows that there was no significant difference in the two groups with regard to fever. Therefore, fever should be removed from the abstract.
  7. Since the authors chose to combine nasal congestion with sore throat, which on its own did not reveal any significant difference between the two groups, an analysis combining headache with body/muscle ache should also be done.
  8. As written, it appears as if the examined breakthrough infections are the Omicron variant. However, this is an assumption as the manuscript doesn’t mention that samples were sequenced. The title, abstract and text should be edited accordingly.

Minor issue

The abstract text ‘In contrast, the frequency of body/muscle aches (22.1% vs. 32.4%, p=0.038) was significantly less among the boosted.’ Might be more clearly written as follows: In contrast, the frequency of body/muscle aches was significantly less among the boosted than non-boosted (22.1% vs. 32.4%, p=0.038).

Reviewer 2 Report

Comments and Suggestions for Authors

Interesting paper from a theoretical and practical perspective using a limited and narrow population at NYMC.

The authors define their objectives and the need for research relative to health effects of vaccinations and booster shots. The paragraph (lns 63-69) is informative, but it should be moved to Methods.  The results claim correlations but not cause and effect which would require a more robust experimental design.

Overall, the authors are commended for presenting a careful description of the study population (Figure 2) and how data were collected and validated.

Figures 1 and 2 are appropriate and present useful information.  Figure 3 appears at the end of the end of the analysis. Adjusting for the time interval is a necessary procedure. It is the most significant result.  It is not clear the actual comparisons relative to the Odd Ratios.  Further, in the discussion section the substantive and statistical significance should be addressed. In light of the data and findings the authors should comment on possible reasons for the non-significant difference of nasal congestion and/or sore throat or body/muscle ache.

Table 3 with its reduced sample size has only one significant finding. As such, I would delete the table and reference the significant finding.  The table does not add much to the analysis.

The discussion might also review Table 2 and provide insight as to explanations as to why some the clinical symptoms were not significant. Perhaps the literature would offer some insights.  Limitations of the study should include comments on external validity and issues with the study population composition since students are more frequently represented than employees.

Round 2

Reviewer 2 Report

Comments and Suggestions for Authors

Response to my previous comments is appreciated.  The authors provide additional useful details.  While I would still prefer that Table 3 be deleted, the authors' preference in this regard is understandable.